# The Effect of Child Abuse and Neglect on Trajectories of Depressive Symptoms and Aggression in Korean Adolescents: Exploring Gender Differences

**DOI:** 10.3390/ijerph19106160

**Published:** 2022-05-19

**Authors:** Chung Choe, Seunghee Yu

**Affiliations:** 1Department of Economics, Konkuk University, Seoul 05029, Korea; choechung@gmail.com; 2Department of Social Welfare, Sungkyul University, Anyang 14097, Korea

**Keywords:** child abuse, child neglect, depressive symptoms, aggression, adolescents

## Abstract

We analyzed gender differences in the effects of child abuse and neglect experienced during adolescence on depressive symptoms and aggression in Korean adolescents using a representative sample of participants over a three-year period. We applied a latent growth model to a sample of 1797 adolescents aged 14–16 from the Korean Children and Youth Panel Survey. Our findings revealed that abuse increased depressive symptoms in early adolescence, while lowering the rate of increase in depressive symptoms over time. Neglect adversely affected depressive symptoms in boys, but not in girls. Abuse increased the initial value of aggression in girls more than in boys, but reduced the increase rate of aggression over time in girls. Neglect increased the initial value of aggression only in boys. Consequently, abuse and neglect experienced during adolescence can affect depressive symptoms and aggression in the individual differently, depending on gender. This study suggests that, in order to reduce depressive symptoms and aggression in adolescents, work should be undertaken to solve the problems of abuse and neglect, and different approaches should be taken according to the gender of the individual.

## 1. Introduction

Depressive symptoms and aggression are typical internalizing and externalizing behavioral problems that can appear in adolescence [1]. Emotional confusion and instability due to physical, hormonal, social, emotional, and psychological changes in the pubertal stage can trigger depressive symptoms or aggressive behavior [2,3]. Adolescents in Asian countries tend to have a higher prevalence of depressive symptoms than those in Western countries [4,5,6]. For example, Steptoe [5] found that around 38% of adolescents in Asian regions, including Korea, presented depressive symptoms, while less than 20% of adolescents in Western countries did so. During adolescence, depression rates rise dramatically, and the gender difference in depression emerges: girls present more depressive symptoms than boys [7,8,9].

Aggression is an action that intentionally causes pain or injury to others through physical or verbal behavior [10]. Adolescence is a period of a high risk of aggression: a sharp increase in antisocial behavior occurs between the ages of 7 and 17, and prevalence and incidence of aggression peaks at age 17 [11]. When children enter adolescence, they begin to admire aggressive peers and regard good students as less attractive [12]. Aggressive behavior provides them with a heightened sense of power and independence and neutralizes feelings of inadequacy that often exist during adolescence [13]. Findings from research conducted over the past few decades indicate that boys demonstrate more physical and verbal aggression than girls, and that such gender differences in aggressive behavior are more pronounced during adolescence [14,15].

Adolescent depressive symptoms and aggression are associated with negative long-term functional and psychiatric outcomes, such as impairments in school performance and interpersonal relationships, criminal behavior, and suicide [16,17]. Accordingly, scholars have been searching for ways to reduce depressive symptoms and aggression in adolescents by finding factors that affect them. Several studies have shown that child abuse and neglect are the main factors that increase depressive symptoms and aggression [18,19]. Child abuse refers to a parenting style in which parents hurt their children physically or emotionally through excessive corporal punishment or harsh language [20]. Child neglect is a pattern of failing to provide for a child’s basic needs, which means that, unlike abuse, parents do not deliver even essential punishment or admonition, are indifferent to the child’s growth and development, and do not provide the child with needed protection and support [21].

Stuewig and McCloskey [22] found that harsh parenting in childhood (9 years old) increases parental rejection in adolescence (15 years old) and leads to an increase in depression in late adolescence (17 years old). Kim and Cicchetti [23] investigated the developmental trajectories of depressive symptoms during elementary school years (6–11 years old) among maltreated and non-maltreated children. It appeared that depressive symptoms decreased over time among non-maltreated and maltreated children. Multivariate growth modeling indicated that, regardless of gender, physical abuse and physical neglect increased initial levels of depressive symptoms. Adults severely abused or neglected as children tend to have difficulties in emotional regulation and forming intimate relationships with others [24]. Abused children between the ages of 6 and 12 are less likely to understand their emotions, express them appropriately, and are at an increased risk of engaging in aggressive behavior [25].

Existing studies have mainly analyzed the effects of abuse experienced in childhood that emerged in childhood, adolescence or adulthood [22,23,24,25,26,27]. However, studies on the association between abuse and neglect experienced in adolescence and the risk for depressive symptoms and aggression are scarce. Abuse and neglect occurring in adolescence tend to be overlooked compared to those which occur during earlier childhood. However, the incidence of abuse and neglect among adolescents is not low, for example, 35.3% of Korean adolescents aged 13–17 experienced abuse or neglect [28]. Because adolescents and children have different physical, cognitive, and social characteristics, experiences and responses to abuse and neglect in adolescence may be different from those of childhood.

There is a differential relationship between maltreatment experiences and depressive symptoms and aggression in boys and girls [23,29,30]. Evans et al. [31] found that the effect sizes of externalizing behavioral problems were significantly larger for boys who experienced domestic violence than for girls. Dodge et al. [32] found that physically abused boys and girls exhibit comparable levels of externalizing behavioral problems, but girls present more internalizing behavioral difficulties. Herrera and McCloskey [33] found that girls experiencing child abuse are more likely than boys to be arrested for violent offenses. Girls exposed to domestic violence were at higher risk than boys for externalizing and internalizing behavioral problems [34].

As such, the precise manifestation of gender differences in the effects of abuse and neglect on depressive symptoms and aggression in adolescents is not agreed upon [31,32,33,34]. This is likely because the relationship between the variables may differ depending on the country and culture that a given study is targeting. Additionally, since the previous studies mainly performed cross-sectional analysis, there would be limitations in revealing the causal relationships between variables. Therefore, we intend to examine the gender differences in the effects of abuse and neglect experienced during adolescence on the changes in depressive symptoms and aggression in Korean adolescents using a representative sample over a three-year period.

## 2. Materials and Methods

### 2.1. Data and Participants

This study used longitudinal data from the Korean Children and Youth Panel Survey (KCYPS)—the fourth-grader panel—conducted by the National Youth Policy Institute in Korea. The KCYPS is an annual survey that collected information on adolescents’ activities, behaviors, and psychological and social characteristics from 2010 to 2016 [35]. The KCYPS started with students in the fourth grade of elementary schools in 2010 and ran a follow-up survey on the same subjects every year until 2016. Samples were extracted using multi-stage stratified cluster sampling. In total, 2378 students enrolled in the fourth-grade of elementary school in 2010 (wave 1) were selected as the original sample. We used data from wave 5 (eighth-graders in 2014) to wave 7 (tenth-graders in 2016). Individuals with missing values were excluded from the sample. The final sample consisted of 1797 Korean adolescents who completed all three waves of the survey. The anonymity and privacy of the participants were ensured during data collection. Because we used secondary data, the approval of the Research Ethics Committee was not required.

### 2.2. Measures

Depressive symptoms and aggression were dependent variables. Depressive symptoms were measured using ten items from the Depression Scales of the Korean Mental Diagnosis Test [36]. Aggression was measured using six items from the Aggressive Behavior Rating Scale [37]. Responses to each question were rated on a 4-point Likert scale, ranging from 1 (*strongly disagree*) to 4 (*strongly agree*). A higher score indicated a higher level of depressive symptoms or aggression. The Cronbach alpha coefficients of the items for depressive symptoms were 0.902 (wave 5), 0.892 (wave 6), 0.893 (wave 7), and for aggression: 0.816 (wave 5), 0.811 (wave 6), 0.823 (wave 7). Independent variables were abuse and neglect, which were measured with four items (α = 0.837 in wave 5) from the Abusive Parenting Scale and four items (α = 0.732 in wave 5) from the Neglectful Parenting Scale, respectively, rated on a 4-point Likert scale [38]. The higher the value, the higher the level of abuse or neglect.

We included household income, health status, satisfaction with academic performance, and peer relationships as confounders, since the prior literature showed a relationship between socio-demographic characteristics and psychological and behavioral problems in adolescents [39,40]. Annual household income was assessed by ranking from “*10 million won or less* = 1”, “*between 10 million and 20 million won* = 2”, to “*up to over 100 million won* = 11”, with 11 classifications in total. Health status was measured with the response to the question, “How do you feel about your health compared to your peers?” on a 4-point Likert scale (“*very unhealthy*” = 1 to “*very healthy*” = 4). Satisfaction with academic performance was measured with the response to the question, “How satisfied are you with your academic performance?” on a 4-point Likert scale (“*very unsatisfied*” = 1 to “*very satisfied*” = 4). Peer relationships were assessed with five items from the School Life Adaptation Scale–Peer Relationships responding on a 4-point Likert scale [41]. The higher the value, the better the peer relationships. Gender was coded as “*male* = 0”, and “*female* = 1”.

### 2.3. Data Analysis

We calculated the mean values of depressive symptoms, aggression, abuse, neglect, and socio-demographic status (household income, health status, satisfaction with academic performance, peer relationships, and age) of the study subjects through descriptive statistics. In addition, a *t*-test was conducted to determine if there is a significant difference between the means of male and female group. We used the latent growth model (LGM) to examine the trajectories of depressive symptoms and aggression. The LGM estimates the magnitude of change at the group and individual levels using longitudinal data. In the first step, we conducted a conditional LGM to examine the effects of abuse and neglect on the trajectories of depressive symptoms and aggression. In the second step, we included socio-demographic variables in the analysis to control their effects on depressive symptoms and aggression. We used data from the fifth wave as explanatory variables, assuming that they were time-invariant covariates [42]. Next, we conducted a multi-group comparison analysis to investigate gender differences in the association between abuse, neglect, depressive symptoms, and aggression. We used the statistical software packages AMOS 23 and SPSS 22 for the application of the LGM.

## 3. Results

### 3.1. Descriptive Statistics

The distributions of the variables used in the LGM model were normal, with skewness values under 2 and kurtosis values under 7 [43]. As shown in Table 1, the level of depressive symptoms was higher in girls than in boys, and it tended to increase over time. The level of reported aggression tended to decrease over time in both boys and girls. Note that in ninth grade (wave 6) and tenth grade (wave 7), the girls’ reported level of aggression was higher than the boys’. We also observed that the prevalence of abuse was higher against boys than girls, and that there were not any significant gender differences in who was subject to neglect. The average annual household income was approximately 45 million won. Boys’ health status and satisfaction with academic performance were higher than those of girls. Average age of the subjects was 13.95 (SD = 0.24), 14.95 (SD = 0.24), and 15.95 (SD = 0.24) in wave 5–7, respectively.

### 3.2. Gender Differences and Association between Abuse, Neglect, Depressive Symptoms, and Aggression

We conducted a conditional LGM to identify how abuse and neglect influence the trajectories of depressive symptoms and aggression. Table 2 reports the results of conditional LGM of depressive symptoms. In step 2, the higher the reported level of abuse, the higher the intercept of depressive symptoms for both boys (β = 0.187, *p* < 0.001) and girls (β = 0.218, *p* < 0.001). Abuse decreased the depressive symptom slope for both boys (β = –0.055, *p* < 0.001) and girls (β = –0.082, *p* < 0.001). These results indicate that adolescents experiencing more significant abuse tend to have higher initial levels of depressive symptoms, but that the increase in depressive symptoms over the years was lower compared to those individuals who experienced less abuse. The change in the overall depressive symptom value indicates that abuse increases depressive symptoms in adolescence. The reason for this increase is likely that abuse has a relatively large impact on the initial value of depressive symptoms, but its effect on the slope is relatively limited.

The better the health status, the lower the initial value of depressive symptoms, and this effect was greater in girls (β = −0.245, *p* < 0.001) than in boys (β = −0.139, *p* < 0.001). The higher the satisfaction with academic performance, the lower the initial value of depressive symptoms in both boys (β = −0.092, *p* < 0.001) and girls (β = −0.107, *p* < 0.001). The better the peer relationships, the lower the initial value of depressive symptoms for both boys (β = −0.301, *p* < 0.001) and girls (β = −0.421, *p* < 0.001). Peer relationships increased the depressive symptoms slope for girls (β = 0.1, *p* < 0.001), but it did not affect the depressive symptoms slope for boys.

Table 3 represents the results of conditional LGM of aggression. In step 2, the greater the level of abuse, the greater the intercept of aggression. Additionally, abuse had a stronger influence on the intercept of girls (β = 0.214, *p* < 0.001) than boys (β = 0.137, *p* < 0.001). Abuse decreased the aggression slope for girls (β = −0.058, *p* < 0.001), but it did not affect the aggression slope for boys. This finding suggests that the decrease in aggression over the years was larger in girls experiencing abuse than those experiencing less abuse. Neglect increased the intercept of aggression in boys (β = 0.083, *p* < 0.01), but it did not affect this intercept of girls. Figure 1 and Figure 2 describe gender differences in the effects of abuse and neglect on depressive symptoms and aggression, respectively. As the satisfaction with academic performance increased, aggression decreased in both boys (β = −0.057, *p* < 0.01) and girls (β = −0.056, *p* < 0.05). The better the peer relationships, the lower the aggression for both boys (β = −0.399, *p* < 0.001) and girls (β = −0.505, *p* < 0.001).

## 4. Discussion

This study investigated the effects of abuse and neglect experienced during adolescence on depressive symptoms and aggression in Korean adolescents, their change over time, and the gender differences between these variables. While previous studies have mainly analyzed the long-term effects that abuse and neglect that occurred in childhood could have on depressive symptoms and aggression in adolescence or adulthood, this study analyzed the changes in depressive symptoms and aggression in adolescents, focusing on abuse and neglect experienced in adolescence.

Abuse increased the initial value of depressive symptoms in adolescence, while lowering the increase rate (slope) of depressive symptoms over time. Abuse had a large effect on depressive symptoms in early adolescence, but the effect gradually decreased, likely because of the increase in adolescents’ adaptability and ability to cope with the abuse over time [44]. However, because the effect of abuse on the slope of depressive symptoms is relatively small compared to the impact of abuse on the intercept of depressive symptoms, adolescents who experience abuse have higher levels of depressive symptoms than those who do not. Consistent with the findings of Kim and Cicchetti [23], who showed that child abuse increases depression in children regardless of gender, abuse experienced in adolescence in this study increased depressive symptoms in both male and female adolescents equally.

Neglect increased the initial level of depressive symptoms only in boys. This is inconsistent with previous research findings that child neglect increases the initial value of depression in children equally in males and females [23]. These differences may be related to the socialization process that occurs before adolescence. Neglect is primarily defined as not providing for a child’s basic requirements, such as food, clothing, and cleanliness. In Korea, gender roles in the family tend to be strictly defined and separated. Because caregiving is primarily viewed as the role of women, girls may more actively learn and internalize caregiving skills from observing their mothers at home. Therefore, during adolescence, girls may have a better ability to take care of themselves than boys. Even if they experience neglect, they may experience less discomfort than boys. However, because boys are more immature than girls in terms of taking care of themselves, they may have more difficulty dealing with neglect, which could lead to increased depressive symptoms.

Abuse increased the initial value of aggression and influenced girls more, but decreased the increase rate (slope) of aggression over time in girls. In the case of experiencing the same abuse, girls have higher aggression than boys, but over time, the aggression increase rate is smaller than that of boys. The effect of abuse on aggression in girls is large in the beginning, but the effect gradually decreases, likely because of the increase in the girl’s ability to adapt and cope with abuse over time [44]. The difference in the initial values of aggression between the two groups is larger than the difference in the slopes. These results support previous studies in which maltreatment has a more significant impact on aggression in girls than in boys [33,34]. These results may be associated with girls’ more interpersonally oriented characteristics than boys in adolescence [45]. During this period, girls especially feel a strong attachment and intimacy with their mothers. Thus, abuse during adolescence from their mothers is likely to cause even greater frustration and anger in girls than in boys, leading to aggressive behaviors [46].

Neglect increased the initial value of aggression only in boys. Parents tend to be more tolerant of aggressive behavior in boys and to dependent behavior in girls [47]. While female adolescents are socialized to be relationship-oriented and to communicate about their emotions more readily, this is less emphasized in male adolescents [48]. While female adolescents value acceptance and intimacy within the group, male adolescents are more interested in acquiring higher status than in being accepted by the group [49]. Therefore, when boys experience stress due to neglect, they are more likely than girls to express it through aggressive behavior [50,51].

Health status reduced depressive symptoms more in girls than in boys. The health-related differences between the genders are remarkable in adolescence [52]. For example, with the onset of puberty, girls are more dissatisfied with their health and receive medical care more often; girls experience more psychosomatic disorders than boys. Therefore, physical health can be more closely related to the depressive symptoms of girls. Peer relationships decreased the initial values of depressive symptoms in both boys and girls. They increased the increase rate of depressive symptoms over time (slope) in girls, but did not affect the boys’ slope. Girls tend to be more interpersonally oriented than boys in adolescence [45]. Good friendships increase a girl’s depressive symptoms in early adolescence. However, over time they can compare their appearance, abilities, and home environments to their peers. This social comparison may occur more frequently among close friends, and it may lead to the increase in depressive symptoms over time [53].

This study suggests that, in order to reduce depressive symptoms and aggression in Korean adolescents, different approaches should be taken according to gender when intervening in situations of abuse and neglect. First, boys who experience neglect exhibit more depressive symptoms and aggression than those who do not. These results suggest that when treating male adolescents with depressive symptoms and aggression problems, therapists should intervene and determine whether the parenting method is neglectful. Furthermore, the socialization process should ensure that boys are being better equipped to take care of themselves than they are currently. Instead of expressing the stress caused by neglect through aggressive behavior, boys should develop the ability to cope with stress in a healthy way, such as through acceptable communication.

Second, adolescents who experience abuse exhibit more aggression than those who do not. Abuse increases girls’ aggression more than boys in early adolescence. These results suggest that when treating adolescents with aggression problems, therapists should intervene and determine whether the parenting method is abusive. This intervention is especially imperative in the case of girls in early adolescence.

Lastly, in order to prevent abuse and neglect of adolescents, it is necessary to educate parents on proper parenting methods. Some parents may think that corporal punishment or harsh scolding of their children sets them on the right path. Other parents may believe that not intervening in their children’s lives fosters their child’s autonomy. Therefore, it is imperative to teach parents about the concepts, scope, and specific examples of abusive or neglectful parenting, and to show them the adverse effects that abuse and neglect can have on their children in adolescence.

### Limitations and Future Research

Our research has some limitations. First, we measured child abuse and neglect during adolescence using self-reported questionnaires. This type of measurement could result in reporting bias. For instance, the subjective criteria for aggression among boys and girls may differ. Boys could underestimate their aggression because verbally and physically aggressive behaviors are more common in their peer groups than in girls’ groups. On the other hand, girls could overestimate their aggression for the same reason. Future studies should collect data through both subjective and objective methods.

Second, there may be gender differences in the type of aggressive behaviors exhibited, but this study did not distinguish between aggression types. In future studies, it will be necessary to compare gender differences in levels of aggression by classifying it into direct, overt aggression, such as physical and verbal attacks, and indirect, relational aggression, such as ostracizing and bullying [54,55].

Third, although the types of child abuse and neglect experienced by each gender may differ, this study could not make those distinctions. A future study could analyze the effects and gender differences of different types of abuse and neglect on depressive symptoms and aggression by classifying abuse into physical, emotional, and sexual abuse, and neglect into physical and emotional neglect [56].

## 5. Conclusions

Although child abuse and neglect occur most frequently in early childhood, abuse and neglect of adolescents is still important, because the incidence is not low and such abuse adversely affects adolescents’ depressive symptoms and aggression. Considering that the socialization process is different for males and females and given that the physical and emotional gender differences increase during the pubertal stage, we hypothesized that abuse and neglect experienced during adolescence may affect depressive symptoms and aggression differently depending on gender. In this study, we found that abuse increased depressive symptoms in adolescents regardless of gender, but neglect only adversely affected depressive symptoms in boys. Abuse increased aggression more in girls than in boys in early adolescence, and neglect only had a negative effect on aggression in boys. These results provide support for gender differences in the effects of abuse and neglect experienced during adolescence on depressive symptoms and aggression. In addition, this study suggests that it is necessary to reduce the depressive symptoms and aggression of adolescents by solving the problems of abuse and neglect, and different approaches according to gender are required.

## Figures and Tables

**Figure 1 ijerph-19-06160-f001:**
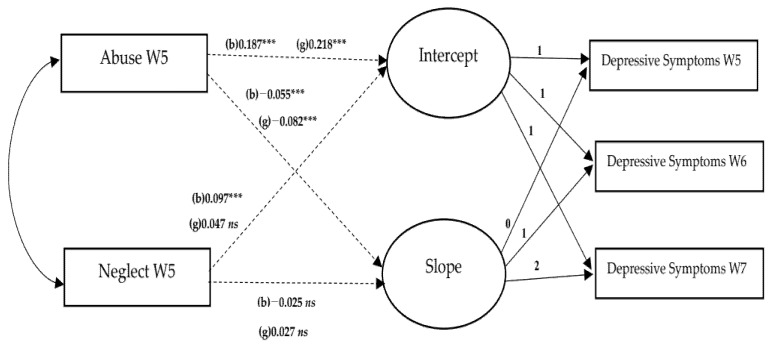
Gender differences in effects of abuse and neglect on depressive symptoms. Note. W5 = wave 5, W6 = wave 6, W7 = wave 7, b = boys, g = girls, *ns* = not significant; *** *p* < 0.001. Covariates (household income, health status, satisfaction with academic performance, and peer relationships in wave 5) were controlled. Dotted lines mean gender differences in effects are not significant at α = 0.05.

**Figure 2 ijerph-19-06160-f002:**
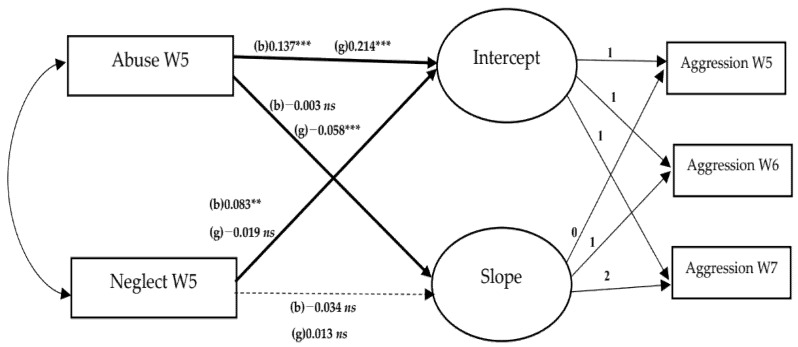
Gender differences in effects of abuse and neglect on aggression. Note. W5 = wave 5, W6 = wave 6, W7 = wave 7, b = boys, g = girls, *ns* = not significant; ** *p* < 0.01, *** *p* < 0.001. Covariates (household income, health status, satisfaction with academic performance, and peer relationships in wave 5) were controlled. Bold lines indicate gender differences in effects are significant at α = 0.05. Dotted lines indicate gender differences in effects are not significant at α = 0.05.

**Table 1 ijerph-19-06160-t001:** Descriptive statistics.

	Overall		Boys		Girls		*t*
Construct	M	STD	M	STD	M	STD	
Depressive symptoms W5	1.77	0.57	1.70	0.55	1.84	0.58	−4.99 ***
Depressive symptoms W6	1.77	0.55	1.70	0.53	1.85	0.55	−5.92 ***
Depressive symptoms W7	1.79	0.55	1.69	0.54	1.89	0.55	−7.58 ***
Aggression W5	1.94	0.56	1.93	0.56	1.96	0.56	−1.00
Aggression W6	1.92	0.54	1.90	0.54	1.95	0.54	−2.03 *
Aggression W7	1.83	0.54	1.79	0.53	1.87	0.54	−3.30 **
Abuse W5	1.62	0.61	1.70	0.65	1.54	0.56	5.60 ***
Neglect W5	1.78	0.55	1.78	0.54	1.78	0.55	−0.17
Household income W5	4.79	2.05	4.74	2.00	4.85	2.10	−1.09
Health status W5	3.32	0.56	3.36	0.57	3.27	0.55	3.33 **
Satisfaction with academic performance W5	2.56	0.78	2.59	0.80	2.52	0.75	1.99 *
Peer relationships W5	3.14	0.41	3.41	0.56	3.30	0.58	−0.02
Age W5	13.95	0.24	13.95	0.24	13.94	0.24	0.75
Age W6	14.95	0.24	14.95	0.24	14.94	0.24	0.75
Age W7	15.95	0.24	15.95	0.24	15.94	0.24	0.75
N	1797		942		855		

Note. W5: Wave 5, W6: Wave 6, W7: Wave 7; * *p* < 0.05, ** *p* < 0.01, *** *p* < 0.001.

**Table 2 ijerph-19-06160-t002:** Conditional LGM estimates of depressive symptoms by gender.

Path	Boys (*N* = 942 for Each Wave)		Girls (*N* = 855 for Each Wave)		C.R
	β	SE	β	SE	
**Step 1**					
Abuse→ICEPT	0.215 ***	0.025	0.283 ***	0.033	1.649
Abuse→SLOPE	−0.057 ***	0.015	−0.093 ***	0.017	−1.61
Neglect→ICEPT	0.207 ***	0.03	0.219 ***	0.034	0.275
Neglect→SLOPE	−0.033	0.018	−0.006	0.018	1.079
**Step 2**					
Abuse→ICEPT	0.187 ***	0.023	0.218 ***	0.029	0.843
Abuse→SLOPE	−0.055 ***	0.015	−0.082 ***	0.017	−1.173
Neglect→ICEPT	0.097 ***	0.03	0.047	0.032	−1.148
Neglect→SLOPE	−0.025	0.019	0.027	0.019	1.954
Household income→ICEPT	0.014	0.007	−0.001	0.008	−1.424
Household income→SLOPE	−0.004	0.005	−0.003	0.005	0.157
Health status→ICEPT	−0.139 ***	0.026	−0.245 ***	0.03	−2.617
Health status→SLOPE	0.018	0.017	0.045 *	0.018	1.14
Satisfaction of academic performance→ICEPT	−0.092 ***	0.019	−0.107 ***	0.022	−0.509
Satisfaction of academic performance→SLOPE	0.012	0.012	0.004	0.013	−0.488
Peer relationships→ICEPT	−0.301 ***	0.038	−0.421 ***	0.048	−1.953
Peer relationships→SLOPE	0.019	0.024	0.1 ***	0.028	2.195

Note. Unconstrained model fit of depressive symptoms (Step 1): 𝒳_2_(6) = 7.085 (*p* = 0.313), TLI = 0.998, CFI = 0.999, RMSEA = 0.010; Unconstrained model fit of depressive symptoms (Step 2): 𝒳_2_(14) = 16.709 (*p* = 0.272), TLI = 0.995, CFI = 0.999, RMSEA = 0.010; Critical ratio (C.R.) outside the range of ±1.96 means gender differences are significant at α = 0.05; * *p* < 0.05, *** *p* < 0.001.

**Table 3 ijerph-19-06160-t003:** Conditional LGM estimates of aggression by gender.

Path	Boys (*N* = 942 for Each Wave)		Girls (*N* = 855 for Each Wave)		C.R
	β	SE	β	SE	
**Step 1**					
Abuse→ICEPT	0.174 ***	0.026	0.272 ***	0.031	2.419
Abuse→SLOPE	−0.006	0.015	−0.063 ***	0.017	−2.529
Neglect→ICEPT	0.202 ***	0.031	0.129 ***	0.032	−1.649
Neglect→SLOPE	−0.045 *	0.018	−0.001	0.018	1.745
**Step 2**					
Abuse→ICEPT	0.137 ***	0.025	0.214 ***	0.029	2.001
Abuse→SLOPE	−0.003	0.015	−0.058 ***	0.017	−2.405
Neglect→ICEPT	0.083 **	0.031	−0.019	0.032	−2.286
Neglect→SLOPE	−0.034	0.019	0.013	0.019	1.743
Household income→ICEPT	0.000	0.008	−0.01	0.008	−0.845
Household income→SLOPE	−0.003	0.005	−0.003	0.005	−0.057
Health status→ICEPT	−0.017	0.028	−0.047	0.03	−0.713
Health status→SLOPE	0.001	0.017	−0.001	0.018	−0.055
Satisfaction of academic performance→ICEPT	−0.057 **	0.02	−0.056 *	0.022	0.055
Satisfaction of academic performance→SLOPE	0.002	0.012	−0.001	0.013	−0.19
Peer relationships→ICEPT	−0.399 ***	0.04	−0.505 ***	0.048	−1.692
Peer relationships→SLOPE	0.042	0.025	0.063 *	0.028	0.56

Note. Unconstrained model fit of aggression (Step 1): 𝒳_2_(6) = 23.659 (*p* = 0.001), TLI = 0.960, CFI = 0.988, RMSEA = 0.040; Unconstrained model fit of aggression (Step 2): 𝒳_2_(14) = 43.609 (*p* < 0.001), TLI = 0.935, CFI = 0.987, RMSEA = 0.034; Critical ratio (C.R.) outside the range of ±1.96 means gender differences are significant at α = 0.05; * *p* < 0.05, ** *p* < 0.01, *** *p* < 0.001.

## Data Availability

Not applicable.

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
