# Peer review of "The Effect of Child Abuse and Neglect on Trajectories of Depressive Symptoms and Aggression in Korean Adolescents: Exploring Gender Differences"

_ijerph, 2022, doi:10.3390/ijerph19106160_

Round 1

Reviewer 1 Report

The introduction part is too long. Authors should summaries it in approximately 300 to 500 words. Avoid presenting lots of figures and comparisons. In this section, I strongly recommend authors  highlight the rate of depression in the adolescence (worldwide and Korea). Relationship between abuse and neglect behavior and depression in the adolescence as well as the importance of reporting this relationship.  The controversy between national studies on this subject and finally why this study is important and how is able to add something on science.

Method: avoid to describe KCYPS extensively, refer  briefly to the main study with reference.

How you determined the sample size?

How you deal with missing data.

How did you checked the normality of data?

Results: Is too long. Authors should avoid repeating tables in the text again. Please briefly highlight just very important results.

Discussion:

Please start discussion with your main findings.

Please highlight your findings at national level.

Author Response

1.The introduction part is too long. Authors should summaries it in approximately 300 to 500 words. Avoid presenting lots of figures and comparisons. In this section, I strongly recommend authors  highlight the rate of depression in the adolescence (worldwide and Korea). Relationship between abuse and neglect behavior and depression in the adolescence as well as the importance of reporting this relationship.  The controversy between national studies on this subject and finally why this study is important and how is able to add something on science.

Response

We thank the referee for the helpful comment which considerably enhances the contents of the paper. In response to your comment, we summarized introduction part and reduced figures and comparisons. We highlighted the rate of depressive symptoms and aggression, relationships between abuse, neglect, depressive symptoms and aggression in adolescence. We stated the controversy in previous studies and clarified the added-value and purpose of the study as below:

 (Introduction)

Depressive symptoms and aggression are typical internalizing and externalizing behavioral problems that can appear in adolescence [1]. Emotional confusion and instability due to physical, hormonal, social, emotional, and psychological changes in the pubertal stage can trigger depressive symptoms or aggressive behavior [2,3]. Adolescents in Asian countries tend to have a higher prevalence of depressive symptoms than those in western countries [4- 6]. For example, Steptoe [5] found that around 38% of adolescents in Asian regions, including Korea, presented depressive symptoms, while less than 20% of adolescents in western countries did so. During adolescence, depression rates rise dramatically and the gender difference in depression emerges: girls present more depressive symptoms than boys [7-9].

Aggression is an action that intentionally causes pain or injury to others through physical or verbal behavior [10]. Adolescence is a period of high risk of aggression: a sharp increase in antisocial behavior occurs between the ages of seven and 17, and prevalence and incidence of aggression peaks at age 17 [11]. When children enter adolescence, they begin to admire aggressive peers and regard good students as less attractive [12]. Aggressive behavior provides them with a heightened sense of power and independence and neutralizes feelings of inadequacy that often exist during adolescence [13]. Findings from research conducted over the past few decades indicate that boys demonstrate more physical and verbal aggression than girls, and that such gender differences in aggressive behavior are more pronounced during adolescence [14,15].

Adolescent depressive symptoms and aggression are associated with negative long-term functional and psychiatric outcomes, such as impairment in school performance and interpersonal relationships, criminal behavior, and suicide [16,17]. Accordingly, scholars have been searching for ways to reduce depressive symptoms and aggression in adolescents by finding factors that affect them. Several studies have shown that child abuse and neglect are the main factors that increase depressive symptoms and aggression [18,19]. Child abuse refers to a parenting style in which parents hurt their children physically or emotionally through excessive corporal punishment or harsh language [20]. Child neglect is a pattern of failing to provide for a child’s basic needs, which means that, unlike abuse, parents do not deliver even essential punishment or admonition, are indifferent to the child’s growth and development, and do not provide the child with needed protection and support [21].

Stuewig and McCloskey [22] found that harsh parenting in childhood increases parental rejection in adolescence and leads to an increase in depression in late adolescence. Kim and Cicchetti [23] investigated the developmental trajectories of depressive symptoms during elementary school years among maltreated and non-maltreated children. It appeared that depressive symptoms decreased over time among non-maltreated and maltreated children. Multivariate growth modeling indicated that, regardless of gender, physical abuse and physical neglect increased initial levels of depressive symptoms. Hostile, neglectful, or inconsistent parenting, usually presented in the families of maltreated children, disrupts the children’s development of emotional regulation skills [24]. Abused children are less likely to understand emotion, express it appropriately, and are at an increased risk of engaging in aggressive behavior [25].

Existing studies mainly analyzed the effects of abuse experienced in childhood that emerged in late adolescence or adulthood [19,22,26,27]. However, studies on the association between abuse and neglect experienced in adolescence and the risk for depressive symptoms and aggression are scarce. Abuse and neglect occurring in adolescence tend to be overlooked compared to those which occur during earlier childhood. However, the incidence of abuse and neglect among adolescents is not low, for example, 35.3% of Korean adolescents aged 13-17 experienced abuse or neglect [28]. Because adolescents and children have different physical, cognitive, and social characteristics, experiences and responses to abuse and neglect in adolescence may be different from those of childhood.

There is a differential relationship between maltreatment experiences and depressive symptoms and aggression in boys and girls [23,29,30]. Evans et al. [31] found that the effect sizes of externalizing behavioral problems were significantly larger for boys who experienced domestic violence than for girls. Dodge et al. [32] found that physically abused boys and girls exhibit comparable levels of externalizing behavioral problems, but girls present more internalizing behavioral difficulties. Herrera and McCloskey [33] found that girls experiencing child abuse are more likely than boys to be arrested for violent offenses. Girls exposed to domestic violence were at higher risk than boys for externalizing and internalizing behavioral problems [34].

As such, the precise manifestation of gender differences in the effects of abuse and neglect on depressive symptoms and aggression in adolescents is not agreed upon [31-34]. This is likely because the relationship between the variables may differ depending on the country and culture that a given study is targeting. Also, since the previous studies mainly performed cross-sectional analysis, there would be limitations in revealing the causal relationships between variables. Therefore, we intend to examine the gender differences in the effects of abuse and neglect experienced during adolescence on the changes in depressive symptoms and aggression in Korean adolescents using a representative sample over a three-year period.

  1. Method: avoid to describe KCYPS extensively, refer briefly to the main study with reference.

Response

In response to your comment, we revised data descriptions as below:

(data and participants)

This study used longitudinal data from the Korean Children and Youth Panel Survey (KCYPS) – the fourth-grader panel – conducted by the National Youth Policy Institute in Korea. The KCYPS started with students in the fourth grade of elementary schools in 2010 and ran a follow-up survey on the same subjects every year until 2016. Samples were extracted using multi-stage stratified cluster sampling. 2,378 students enrolled in the fourth-grade of elementary school in 2010 (wave 1) were selected as the original sample. We used data from wave 5 (eighth-graders in 2014) to wave 7 (tenth-graders in 2016). Individuals with missing values were excluded from the sample. The final sample consisted of 1,797 Korean adolescents who completed all three waves of the survey.

  1. How you determined the sample size?

Response

The population of KCYPS was students enrolled in the 4th grade of elementary school nationwide in 2010, and the 2009 national school statistics from the Ministry of Education were used as a sampling frame. The sample size was determined after consultation with experts, referring to representativeness, precision, research cost, and the retention rate of the original sample of the Korea Youth Penal Survey (KYPS) conducted in the past by the National Youth Policy Institute. The sampling was done using multi-stage stratified cluster sampling.

  1. How you deal with missing data.

Response

Individuals with missing values were excluded from the sample. We stated this in data and participant section.

  1. How did you checked the normality of data?

Response

We checked the normality of data through testing skewness and kurtosis of variables. We stated this in the descriptive statistics section as below:

(descriptive statistics)

The distributions of the variables used in the LGM model were normal, with skewness values under 2 and kurtosis values under 7 [42].

  1. Results: Is too long. Authors should avoid repeating tables in the text again. Please briefly highlight just very important results.

 Response

In response to your comment, we revised the result section briefly highlighting very important results as below:

3.1. Descriptive Statistics

The distributions of the variables used in the LGM model were normal, with skewness values under 2 and kurtosis values under 7 [42]. As shown in Table 1, the level of depressive symptoms was higher in girls than in boys, and it tended to increase over time. The level of reported aggression tended to decrease over time in both boys and girls. Note that in ninth grade (wave 6) and tenth grade (wave 7), the girls’ reported level of aggression was higher than the boys’. We also observed that the prevalence of abuse was higher against boys than girls, and that there were not any significant gender differences in who was subject to neglect. The average annual household income was approximately 45 million won. Boys’ health status and satisfaction with academic performance were higher than those of girls. Average age of the subjects was 13.95 (SD = 0.24), 14.95 (SD = 0.24), 15.95 (SD = 0.24) in wave 5-7, respectively.

3.2. Gender differences and association between abuse, neglect, depressive symptoms, and aggression

We conducted a conditional LGM to identify how abuse and neglect influence the trajectories of depressive symptoms and aggression. Table 2 reports the results. The higher the reported level of abuse, the higher the intercept of depressive symptoms for both boys (b = 0.187, p < 0.001) and girls (b = 0.218, p < 0.001). Abuse decreased the depressive symptom slope for both boys (b = –0.055, p < 0.001) and girls (b = –0.082, p < 0.001). These results indicate that adolescents experiencing more significant abuse tend to have higher initial levels of depressive symptoms, but that the increase in depressive symptoms over the years was lower compared to those individuals who experienced less abuse. The change in the overall depressive symptom value indicates that abuse increases depressive symptoms in adolescence. The reason for this increase is likely that abuse has a relatively large impact on the initial value of depressive symptoms, but its effect on the slope is relatively limited.

The greater the level of abuse, the greater the intercept of aggression. Additionally, abuse had a stronger influence on the intercept of girls (b = 0.214, p < 0.001) than boys (b = 0.137, p < 0.001). Abuse decreased the aggression slope for girls (b = -0.058, p < 0.001), but it did not affect the aggression slope for boys. This finding suggests that the decrease in aggression over the years was larger in girls experiencing abuse than those experiencing less abuse. Neglect increased the intercept of aggression in boys, but it did not affect the intercept of girls. Figure 1 and 2 describe gender differences in the effects of abuse and neglect on depressive symptoms and aggression, respectively.

  1. Discussion:

Please start discussion with your main findings.

Please highlight your findings at national level.

Response

In response to your comment, we revised the discussion section starting with main findings and highlighting findings at national level as below:

(discussion)

This study investigated the effects of abuse and neglect experienced during adolescence on depressive symptoms and aggression in Korean adolescents, their change over time, and the gender differences between these variables. Abuse increased initial levels of depressive symptoms while decreasing the depressive symptom slope in both boys and girls. Abuse had a large effect on depressive symptoms in early adolescence, but the effect gradually decreased, likely because of the increase in adolescents' adaptability and ability to cope with the abuse over time [43]. However, because the effect of abuse on the slope of depressive symptoms is relatively small compared to the impact of child abuse on the intercept of depressive symptoms, adolescents who experience child abuse have higher levels of depressive symptoms than those who do not. Consistent with the findings of Kim and Cicchetti [23], who showed that child abuse increases depression in children regardless of gender, adolescent abuse in this study increased depressive symptoms in both Korean males and females equally.

Neglect increased the initial level of depressive symptoms only in boys. This is inconsistent with previous research findings that child neglect increases the initial value of depression in children equally in males and females [23]. These differences may be related to the socialization process that occurs before adolescence. Neglect is primarily defined as not providing for a child's basic requirements, such as food, clothing, and cleanliness. In Korea, gender roles in the family tend to be strictly defined and separated. Because caregiving is primarily viewed as the role of women, girls may more actively learn and internalize caregiving skills from observing their mothers at home. Therefore, during adolescence, girls may have a better ability to take care of themselves than boys. Even if they experience neglect, they may experience less discomfort than boys. However, because boys are more immature than girls in terms of taking care of themselves, they may have more difficulty dealing with neglect, which could lead to increased depressive symptoms.

Abuse increased the initial value of aggression and influenced girls more. It also decreased the slope of aggression in girls. In the case of experiencing the same abuse, girls have higher aggression than boys, but over time the aggression decrease rate is greater than that of boys. The effect of neglect on aggression in girls is large in the beginning, but the effect gradually decreases likely because of the increase in the girl's ability to adapt and cope with abuse over time [43]. The difference in the initial values of aggression between the two groups is larger than the difference in the slopes. These results support previous studies in which maltreatment has a more significant impact on aggression in girls than in boys [33,34]. These results may be associated with girls’ more interpersonal-oriented characteristics than boys in adolescence [44]. During this period, girls especially feel a strong attachment and intimacy with their mothers. Thus, abuse during adolescence from their mothers is likely to cause even greater frustration and anger in girls than in boys, leading to aggressive behaviors [45].

Neglect increased the initial value ​​of aggression only in boys. Parents tend to be more tolerant of aggressive behavior in boys and to dependent behavior in girls [46]. While female adolescents are socialized to be relationship-oriented and to communicate about their emotions more readily, this is less emphasized in male adolescents [47]. While female adolescents value acceptance and intimacy within the group, male adolescents are more interested in acquiring higher status than in being accepted by the group [48]. Therefore, when boys experience stress due to neglect, they are more likely to express it through aggressive behavior than girls [49,50]. 

This study suggests that, in order to reduce depressive symptoms and aggression in Korean adolescents, different approaches should be taken according to gender when intervening in situations of abuse and neglect. First, boys who experience neglect exhibit more depressive symptoms and aggression than those who do not. These results suggest that when treating male adolescents with depressive symptoms and aggression problems, therapists should intervene and determine whether the parenting method is neglectful. Furthermore, the socialization process should ensure that boys are being better equipped to take care of themselves than they are currently. Instead of expressing the stress caused by neglect through aggressive behavior, boys should develop the ability to cope with stress in another, healthier way, such as through acceptable communication.

Second, adolescents who experience abuse exhibit more aggression than those who do not. Adolescent abuse increases girls’ aggression more than boys in early adolescence. These results suggest that when treating adolescents with aggression problems, therapists should intervene and determine whether the parenting method is abusive. This intervention is especially imperative in the case of girls in early adolescence.

Lastly, in order to prevent abuse and neglect of adolescents, it is necessary to educate parents on proper parenting methods. Some parents may think that corporal punishment or harsh scolding of their children sets them on the right path. Other parents may believe that not intervening in their children’s lives fosters their child’s autonomy. Therefore, it is imperative to teach parents about the concepts, scope, and specific examples of abusive or neglectful parenting, and to show them the adverse effects that abuse and neglect can have on their children in adolescence.

Reviewer 2 Report

Thank you for having the opportunity to read this interesting manuscript. The manuscript is well written and the study is interesting. I have some suggestions and comments that I hope will help the authors to enhance the readability and understanding of the study and thus the impact of the manuscript.

Abstract.

The abstract should be more informative. From the begin we understand the longitudinal nature of the study. However, for how long? How old are the participants? State more clearly what this study add?

Introduction

The authors should help the reader to better understand the added-value of the present study. It is not easy to see the differences with the previous studies and what this study bring for new information. For instance, how this study differed from the one of Stuewig & McCloskey? This should be clarified.

The gap in literature or the limitations of the previous studies should be more clearly described in order that the reader understand the usefulness of the current study.

The current study section should be strengthen. The rationale of the study, the hypothesis and the added-value of the current study have to be more clearly stated in this part.

Method

The sample should be described in more details. Could the authors report a socio-demographic table? Testing the gender differences on the socio-demographic characteristic is important in order to assess if covariates should be added in the model.

Could the authors describe the ethical procedure.

Data analyses.

This section should describe all analyses conducted. The analyses should be more clearly related to the aims of the study. Why the authors tested gender differences? This is not clearly stated as an aim. Moreover, I do not understand well the sense of the analyses reported in section 3.2. As the next analysis seem to encompass also the notion of trajectories in symptoms. Perhaps authors should report some results in supplementary file? The main point here, from my point of view, is to related more clearly the statistical analyses and an aim of the study.

Instead of having a figure with the model tested, could the authors report a figure with the results (beta included in the figure) for the main analyses?

Results

The authors stated the prevalence in the introduction. Did the authors could provide the prevalence in their sample. The gap between previous study and this one could be helpful to appreciate one limitation of the study, namely the self-report nature of the data. Thus, the reader might appreciate the over or under-report of abuse and neglect compared to previous data from South Korea.

Add the ages of the Wave. It is difficult to understand the mean age of each wave and thus to appreciate the longitudinal nature of the study.

Discussion

Reference should sustain the statment of the underlying processed related to the decrease of aggression over time.

Child abuse and neglect is differentiated in the introduction and in the results. In the table 4, we observe a differentiated effect between both notions. However, in the discussion the authors did not interpret / discuss this aspect anymore. This should be taken into account and discussed.

A concluding paragraph is lacking.

Author Response

  1. Abstract

The abstract should be more informative. From the begin we understand the longitudinal nature of the study. However, for how long? How old are the participants? State more clearly what this study add?

Response

We thank the referee for the helpful comment which considerably enhances the contents of the paper. In response to your comment, we revised abstract adding longitudinal data period, age of the participants, and added values of the study as below:

(abstract)

Abstract: We analyzed gender differences in the effects of child abuse and neglect experienced during adolescence on depressive symptoms and aggression in Korean adolescents using a representative sample of participants over a three-year period. We applied a latent growth model to a sample of 1,798 adolescents aged 14-16 from the Korean Children and Youth Panel Survey. Our findings revealed that abuse increased the initial levels of depressive symptoms in both boys and girls while decreasing the slope. Neglect adversely affected depressive symptoms in boys, but not in girls. Abuse increased the initial value of aggression in girls more than in boys. It also decreased the slope of aggression in girls. Neglect increased the initial value ​​of aggression only in boys. Consequently, abuse and neglect experienced during adolescence can affect depressive symptoms and aggression in the individual differently depending on gender. This study suggests that, in order to reduce depressive symptoms and aggression in adolescents, work should be done to solve the problems of abuse and neglect, and different approaches should be taken according to the gender of the individual.

  1. Introduction

The authors should help the reader to better understand the added-value of the present study. It is not easy to see the differences with the previous studies and what this study bring for new information. For instance, how this study differed from the one of Stuewig & McCloskey? This should be clarified.

The gap in literature or the limitations of the previous studies should be more clearly described in order that the reader understand the usefulness of the current study.

The current study section should be strengthen. The rationale of the study, the hypothesis and the added-value of the current study have to be more clearly stated in this part.

Response

In response to your comment, we revised introduction clarifying the differences with the previous studies, and the purpose and added values of the study as below:

(Introduction)

Depressive symptoms and aggression are typical internalizing and externalizing behavioral problems that can appear in adolescence [1]. Emotional confusion and instability due to physical, hormonal, social, emotional, and psychological changes in the pubertal stage can trigger depressive symptoms or aggressive behavior [2,3]. Adolescents in Asian countries tend to have a higher prevalence of depressive symptoms than those in western countries [4- 6]. For example, Steptoe [5] found that around 38% of adolescents in Asian regions, including Korea, presented depressive symptoms, while less than 20% of adolescents in western countries did so. During adolescence, depression rates rise dramatically and the gender difference in depression emerges: girls present more depressive symptoms than boys [7-9].

Aggression is an action that intentionally causes pain or injury to others through physical or verbal behavior [10]. Adolescence is a period of high risk of aggression: a sharp increase in antisocial behavior occurs between the ages of seven and 17, and prevalence and incidence of aggression peaks at age 17 [11]. When children enter adolescence, they begin to admire aggressive peers and regard good students as less attractive [12]. Aggressive behavior provides them with a heightened sense of power and independence and neutralizes feelings of inadequacy that often exist during adolescence [13]. Findings from research conducted over the past few decades indicate that boys demonstrate more physical and verbal aggression than girls, and that such gender differences in aggressive behavior are more pronounced during adolescence [14,15].

Adolescent depressive symptoms and aggression are associated with negative long-term functional and psychiatric outcomes, such as impairment in school performance and interpersonal relationships, criminal behavior, and suicide [16,17]. Accordingly, scholars have been searching for ways to reduce depressive symptoms and aggression in adolescents by finding factors that affect them. Several studies have shown that child abuse and neglect are the main factors that increase depressive symptoms and aggression [18,19]. Child abuse refers to a parenting style in which parents hurt their children physically or emotionally through excessive corporal punishment or harsh language [20]. Child neglect is a pattern of failing to provide for a child’s basic needs, which means that, unlike abuse, parents do not deliver even essential punishment or admonition, are indifferent to the child’s growth and development, and do not provide the child with needed protection and support [21].

Stuewig and McCloskey [22] found that harsh parenting in childhood increases parental rejection in adolescence and leads to an increase in depression in late adolescence.  Kim and Cicchetti [23] investigated the developmental trajectories of depressive symptoms during elementary school years among maltreated and non-maltreated children. It appeared that depressive symptoms decreased over time among non-maltreated and maltreated children. Multivariate growth modeling indicated that, regardless of gender, physical abuse and physical neglect increased initial levels of depressive symptoms. Hostile, neglectful, or inconsistent parenting, usually presented in the families of maltreated children, disrupts the children’s development of emotional regulation skills [24]. Abused children are less likely to understand emotion, express it appropriately, and are at an increased risk of engaging in aggressive behavior [25].

Existing studies mainly analyzed the effects of abuse experienced in childhood that emerged in late adolescence or adulthood [19,22,26,27]. However, studies on the association between abuse and neglect experienced in adolescence and the risk for depressive symptoms and aggression are scarce. Abuse and neglect occurring in adolescence tend to be overlooked compared to those which occur during earlier childhood. However, the incidence of abuse and neglect among adolescents is not low, for example, 35.3% of Korean adolescents aged 13-17 experienced abuse or neglect [28]. Because adolescents and children have different physical, cognitive, and social characteristics, experiences and responses to abuse and neglect in adolescence may be different from those of childhood.

There is a differential relationship between maltreatment experiences and depressive symptoms and aggression in boys and girls [23,29,30]. Evans et al. [31] found that the effect sizes of externalizing behavioral problems were significantly larger for boys who experienced domestic violence than for girls. Dodge et al. [32] found that physically abused boys and girls exhibit comparable levels of externalizing behavioral problems, but girls present more internalizing behavioral difficulties. Herrera and McCloskey [33] found that girls experiencing child abuse are more likely than boys to be arrested for violent offenses. Girls exposed to domestic violence were at higher risk than boys for externalizing and internalizing behavioral problems [34].

As such, the precise manifestation of gender differences in the effects of abuse and neglect on depressive symptoms and aggression in adolescents is not agreed upon [31-34]. This is likely because the relationship between the variables may differ depending on the country and culture that a given study is targeting. Also, since the previous studies mainly performed cross-sectional analysis, there would be limitations in revealing the causal relationships between variables. Therefore, we intend to examine the gender differences in the effects of abuse and neglect experienced during adolescence on the changes in depressive symptoms and aggression in Korean adolescents using a representative sample over a three-year period.

  1. Method

3.1. The sample should be described in more details. Could the authors report a socio-demographic table? Testing the gender differences on the socio-demographic characteristic is important in order to assess if covariates should be added in the model.

Response

In response of your comments, we reported a socio-demographic table (Table 1). We tested the gender differences on the socio-demographic characteristics (household income, health status, satisfaction with academic performance, peer relationships). We used those variables as covariates in the model. We stated this in the measures section as below: 

(Measures)

We included household income, health status, satisfaction with academic performance, and peer relationships as confounders since prior literature showed a relationship between socio-demographic characteristics and psychological and behavioral problems in adolescents [38,39]. Annual household income was assessed by ranking from “10 million won or less = 1,” “between 10 million and 20 million won = 2,” to “up to over 100 million won = 11,” with 11 classifications in total. Health status was measured with the response to the question, “How do you feel about your health compared to your peers?” on a 4-point Likert scale (“very unhealthy” = 1 to “very healthy” = 4). Satisfaction with academic performance was measured with the response to the question, “How satisfied are you with your academic performance?” on a 4-point Likert scale (“very unsatisfied” = 1 to “very satisfied” = 4). Peer relationships were assessed with five items from the School Life Adaptation Scale – Peer Relationships responding on a 4-point Likert scale [40]. The higher the value, the better the peer relationships. Gender was coded as “male = 0,” and “female = 1.”

3.2. Could the authors describe the ethical procedure.

Response

We described the ethical procedures in the data and participants section as below:

(data and participants)

The anonymity and privacy of the participants was ensured during data collection. Because we used secondary data, the approval of the Research Ethics Committee was not required.

  1. Data analyses.

4.1. This section should describe all analyses conducted. The analyses should be more clearly related to the aims of the study. Why the authors tested gender differences? This is not clearly stated as an aim. Moreover, I do not understand well the sense of the analyses reported in section 3.2. As the next analysis seem to encompass also the notion of trajectories in symptoms. Perhaps authors should report some results in supplementary file? The main point here, from my point of view, is to related more clearly the statistical analyses and an aim of the study.

Response

In response to your comments, we removed Table 2 and Table 3 that your mentioned, and we focused on the statistical analysis related to an aim of the study, which is in the new Table2.

4.2. Instead of having a figure with the model tested, could the authors report a figure with the results (beta included in the figure) for the main analyses?

Response

In response to your comments, we reported figures with the results for the main analysis (Figure 1 and 2).

  1. Results

5.1. The authors stated the prevalence in the introduction. Did the authors could provide the prevalence in their sample. The gap between previous study and this one could be helpful to appreciate one limitation of the study, namely the self-report nature of the data. Thus, the reader might appreciate the over or under-report of abuse and neglect compared to previous data from South Korea.

Response

The child abuse and neglect Korea report (National Child Protection Agency, 2017) presented the prevalence rate by calculating the percentage of children who experienced abuse and neglect compared to all children, based on the number of reported cases of child abuse and neglect. In contrast, the KCYPS data used in this study rated the degree of abuse and neglect experienced by adolescents on a 4-point Likert scale, ranging from 1 (not at all) to 4 (very abusive/neglectful). Higher scores indicate more severe abuse and neglect. As such, the child abuse and neglect Korea report (National Child Protection Agency, 2017) and KCYPS have different methods of measuring abuse and neglect, so it seems difficult to draw a conclusion by comparing the prevalence rates with these two data.

5.2. Add the ages of the Wave. It is difficult to understand the mean age of each wave and thus to appreciate the longitudinal nature of the study.

Response

In response to your comments, we added average age of each wave in Table 1 and stated it in the descriptive statistics section as below:

(descriptive statistics)

Average age of the subjects was 13.95 (SD = 0.24), 14.95 (SD = 0.24), 15.95 (SD = 0.24) in wave 5-7, respectively.

  1. Discussion

6.1. Reference should sustain the statment of the underlying processed related to the decrease of aggression over time.

Response

In response to your comments, we added references to sustain the statement of the underlying processed related to the decrease of depressive symptoms and aggression over time in the discussion section as below:

(discussion)

This study investigated the effects of abuse and neglect experienced during adolescence on depressive symptoms and aggression in Korean adolescents, their change over time, and the gender differences between these variables. Abuse increased initial levels of depressive symptoms while decreasing the depressive symptom slope in both boys and girls. Abuse had a large effect on depressive symptoms in early adolescence, but the effect gradually decreased, likely because of the increase in adolescents' adaptability and ability to cope with the abuse over time [43]. However, because the effect of abuse on the slope of depressive symptoms is relatively small compared to the impact of child abuse on the intercept of depressive symptoms, adolescents who experience child abuse have higher levels of depressive symptoms than those who do not.

Abuse increased the initial value of aggression and influenced girls more. It also decreased the slope of aggression in girls. In the case of experiencing the same abuse, girls have higher aggression than boys, but over time the aggression decrease rate is greater than that of boys. The effect of neglect on aggression in girls is large in the beginning, but the effect gradually decreases likely because of the increase in the girl's ability to adapt and cope with abuse over time [43]. The difference in the initial values of aggression between the two groups is larger than the difference in the slopes.

6.2. Child abuse and neglect is differentiated in the introduction and in the results. In the table 4, we observe a differentiated effect between both notions. However, in the discussion the authors did not interpret / discuss this aspect anymore. This should be taken into account and discussed.

Response

In response to your comments, we discussed the results distinguishing abuse and neglect as below:

(discussion)

This study investigated the effects of abuse and neglect experienced during adolescence on depressive symptoms and aggression in Korean adolescents, their change over time, and the gender differences between these variables. Abuse increased initial levels of depressive symptoms while decreasing the depressive symptom slope in both boys and girls. Abuse had a large effect on depressive symptoms in early adolescence, but the effect gradually decreased, likely because of the increase in adolescents' adaptability and ability to cope with the abuse over time [43]. However, because the effect of abuse on the slope of depressive symptoms is relatively small compared to the impact of child abuse on the intercept of depressive symptoms, adolescents who experience child abuse have higher levels of depressive symptoms than those who do not. Consistent with the findings of Kim and Cicchetti [23], who showed that child abuse increases depression in children regardless of gender, adolescent abuse in this study increased depressive symptoms in both Korean males and females equally.

Neglect increased the initial level of depressive symptoms only in boys. This is inconsistent with previous research findings that child neglect increases the initial value of depression in children equally in males and females [23]. These differences may be related to the socialization process that occurs before adolescence. Neglect is primarily defined as not providing for a child's basic requirements, such as food, clothing, and cleanliness. In Korea, gender roles in the family tend to be strictly defined and separated. Because caregiving is primarily viewed as the role of women, girls may more actively learn and internalize caregiving skills from observing their mothers at home. Therefore, during adolescence, girls may have a better ability to take care of themselves than boys. Even if they experience neglect, they may experience less discomfort than boys. However, because boys are more immature than girls in terms of taking care of themselves, they may have more difficulty dealing with neglect, which could lead to increased depressive symptoms.

Abuse increased the initial value of aggression and influenced girls more. It also decreased the slope of aggression in girls. In the case of experiencing the same abuse, girls have higher aggression than boys, but over time the aggression decrease rate is greater than that of boys. The effect of neglect on aggression in girls is large in the beginning, but the effect gradually decreases likely because of the increase in the girl's ability to adapt and cope with abuse over time [43]. The difference in the initial values of aggression between the two groups is larger than the difference in the slopes. These results support previous studies in which maltreatment has a more significant impact on aggression in girls than in boys [33,34]. These results may be associated with girls’ more interpersonal-oriented characteristics than boys in adolescence [44]. During this period, girls especially feel a strong attachment and intimacy with their mothers. Thus, abuse during adolescence from their mothers is likely to cause even greater frustration and anger in girls than in boys, leading to aggressive behaviors [45].

Neglect increased the initial value ​​of aggression only in boys. Parents tend to be more tolerant of aggressive behavior in boys and to dependent behavior in girls [46]. While female adolescents are socialized to be relationship-oriented and to communicate about their emotions more readily, this is less emphasized in male adolescents [47]. While female adolescents value acceptance and intimacy within the group, male adolescents are more interested in acquiring higher status than in being accepted by the group [48]. Therefore, when boys experience stress due to neglect, they are more likely to express it through aggressive behavior than girls [49,50]. 

This study suggests that, in order to reduce depressive symptoms and aggression in Korean adolescents, different approaches should be taken according to gender when intervening in situations of abuse and neglect. First, boys who experience neglect exhibit more depressive symptoms and aggression than those who do not. These results suggest that when treating male adolescents with depressive symptoms and aggression problems, therapists should intervene and determine whether the parenting method is neglectful. Furthermore, the socialization process should ensure that boys are being better equipped to take care of themselves than they are currently. Instead of expressing the stress caused by neglect through aggressive behavior, boys should develop the ability to cope with stress in another, healthier way, such as through acceptable communication.

Second, adolescents who experience abuse exhibit more aggression than those who do not. Adolescent abuse increases girls’ aggression more than boys in early adolescence. These results suggest that when treating adolescents with aggression problems, therapists should intervene and determine whether the parenting method is abusive. This intervention is especially imperative in the case of girls in early adolescence.

  1. A concluding paragraph is lacking.

Response

In response to your comments, we added conclusion part as below:

  1. Conclusions

Although child abuse and neglect occurs most frequently in early childhood, abuse and neglect of adolescents is still important, because the incidence is not low and such abuse adversely affects adolescents’ depressive symptoms and aggression. Considering that the socialization process is different for males and females and given that the physical and emotional gender differences increase during the pubertal stage, we hypothesized that abuse and neglect experienced during adolescence may affect depressive symptoms and aggression differently depending on gender. In this study, we found that abuse increased depressive symptoms in adolescents regardless of gender, but neglect only adversely affected depressive symptoms in boys. Abuse increased aggression more in girls than in boys in early adolescence, and neglect only had a negative effect on aggression in boys. These results provide support for gender differences in the effects of abuse and neglect experienced during adolescence on depressive symptoms and aggression. In addition, this study suggests that it is necessary to reduce the depressive symptoms and aggression of adolescents by solving the problems of abuse and neglect, and different approaches according to gender are required.

Round 2

Reviewer 1 Report

Recommended changes have done. 

Author Response

We appreciate your feedback.

Reviewer 2 Report

The authors did an important work on the paper and enhance the readability of the paper. However, from my point of view I have some concerns that have to be addressed and some clarifications that seems needed before I could recommend the manuscript for publication.

An important conceptual point.

The majority of the previous studies on abuse and neglect examined the long term effect of these phenomena which happens during infancy. In this study, the authors examined the impact of these phenomena occurring during adolescence. This main difference should be more stressed. Moreover, in the introduction and discussion section this should be described. For instance, when presenting the previous studies, it should be clear if this notion are studied from infancy or in other period of development.

Abstract

I think it is important to translate the “slope” in terms more understandable for the reader. For instance, what does it mean a decrease of the slope?

Introduction

The comment above about the differentiation of the impact of abuse and neglect in the different period of development should help the authors to affine the introduction.

Method

A finer description of the sample is still needed. The authors should separate the measures which are used as outcomes, to the socio-demographic ones (e.g., household income, satisfaction, age, etc.). Could authors cite a paper where the sample are described in more details.

The data analyses section might report all analyses conducted and explain / relate them to the hypotheses / aims.

Results

First, authors should study the differences in socio-demographic data. Now, there are combined with the outcomes. I think it would add clarity for the reader if the analyses are conducted step by step.

Regarding the amount of comparison conducted, I think the authors should conduct a MANCOVA, and/or correct this analysis for multiple testing.

The Figures should be corrected I guess. The outcomes should be depression W5, but then W6 and W7 (not W5 in each outcomes, not?)

Discussion

In this section, authors should avoid technical terms. Slope should be “translated” in usual words, to help the reader understand more the impact of the results in the understanding of the development or occurrence of psychopathologies during adolescence.

Authors describe all results that they report. For instance, how can we interpret that peer relationships is related to symptoms.

Minor point

Some sections (e.g., 2.1) are in italic which does not have to be the case.

Author Response

  1. An important conceptual point.

The majority of the previous studies on abuse and neglect examined the long term effect of these phenomena which happens during infancy. In this study, the authors examined the impact of these phenomena occurring during adolescence. This main difference should be more stressed. Moreover, in the introduction and discussion section this should be described. For instance, when presenting the previous studies, it should be clear if this notion are studied from infancy or in other period of development.

Response

We thank the referee for the helpful comment which considerably enhances the contents of the paper. In response to your comment, we stressed the main difference between previous studies and this paper in introduction and discussion as below:

p.2) Introduction

Stuewig and McCloskey [22] found that harsh parenting in childhood (9 years old) increases parental rejection in adolescence (15 years old) and leads to an increase in depression in late adolescence (17 years old). Kim and Cicchetti [23] investigated the developmental trajectories of depressive symptoms during elementary school years (6-11 years old) among maltreated and non-maltreated children. It appeared that depressive symptoms decreased over time among non-maltreated and maltreated children. Multivariate growth modeling indicated that, regardless of gender, physical abuse and physical neglect increased initial levels of depressive symptoms. Adults severely abused or neglected as children tend to have difficulties in emotional regulation and forming intimate relationships with others [24]. Abused children between the ages of 6 and 12 are less likely to understand their emotions, express it appropriately, and are at an increased risk of engaging in aggressive behavior [25].

Existing studies mainly analyzed the effects of abuse experienced in childhood that emerged in childhood, adolescence or adulthood [22-27]. However, studies on the association between abuse and neglect experienced in adolescence and the risk for depressive symptoms and aggression are scarce. Abuse and neglect occurring in adolescence tend to be overlooked compared to those which occur during earlier childhood. However, the incidence of abuse and neglect among adolescents is not low, for example, 35.3% of Korean adolescents aged 13-17 experienced abuse or neglect [28]. Because adolescents and children have different physical, cognitive, and social characteristics, experiences and responses to abuse and neglect in adolescence may be different from those of childhood.

p.8) Discussion

This study investigated the effects of abuse and neglect experienced during adolescence on depressive symptoms and aggression in Korean adolescents, their change over time, and the gender differences between these variables. While previous studies mainly analyzed the long term effects that abuse and neglect that occurred in childhood could have on depressive symptoms and aggression in adolescence or adulthood, this study analyzed the changes in depressive symptoms and aggression in adolescents focusing on the abuse and neglect experienced in adolescence.

  1. Abstract

I think it is important to translate the “slope” in terms more understandable for the reader. For instance, what does it mean a decrease of the slope?

Response

In response to your comment, we translated the meaning of slope as below:

p.1) Abstract

Our findings revealed that abuse increased the depressive symptoms in early adolescence, while lowering the rate of increase in depressive symptoms over time. Neglect adversely affected depressive symptoms in boys, but not in girls. Abuse increased the initial value of aggression in girls more than in boys, but reduced the increase rate of aggression over time in girls.

  1. Introduction

The comment above about the differentiation of the impact of abuse and neglect in the different period of development should help the authors to affine the introduction.

Response

In response to your comment, we revised the introduction as below:

p.2) Introduction

Stuewig and McCloskey [22] found that harsh parenting in childhood (9 years old) increases parental rejection in adolescence (15 years old) and leads to an increase in depression in late adolescence (17 years old). Kim and Cicchetti [23] investigated the developmental trajectories of depressive symptoms during elementary school years (6-11 years old) among maltreated and non-maltreated children. It appeared that depressive symptoms decreased over time among non-maltreated and maltreated children. Multivariate growth modeling indicated that, regardless of gender, physical abuse and physical neglect increased initial levels of depressive symptoms. Adults severely abused or neglected as children tend to have difficulties in emotional regulation and forming intimate relationships with others [24]. Abused children between the ages of 6 and 12 are less likely to understand their emotions, express it appropriately, and are at an increased risk of engaging in aggressive behavior [25].

Existing studies mainly analyzed the effects of abuse experienced in childhood that emerged in childhood, adolescence or adulthood [22-27]. However, studies on the association between abuse and neglect experienced in adolescence and the risk for depressive symptoms and aggression are scarce. Abuse and neglect occurring in adolescence tend to be overlooked compared to those which occur during earlier childhood. However, the incidence of abuse and neglect among adolescents is not low, for example, 35.3% of Korean adolescents aged 13-17 experienced abuse or neglect [28]. Because adolescents and children have different physical, cognitive, and social characteristics, experiences and responses to abuse and neglect in adolescence may be different from those of childhood.

  1. Method

A finer description of the sample is still needed. The authors should separate the measures which are used as outcomes, to the socio-demographic ones (e.g., household income, satisfaction, age, etc.). Could authors cite a paper where the sample are described in more details.

Response

In response of your comment, we added a finer description and citation of the sample as below:

p.3)

The KCYPS is an annual survey that collected information on adolescents’ activities, behaviors, and psychological and social characteristics from 2010 to 2016 [35].

[35] National Youth Policy Institute. The Korean Children and Youth Panel Survey User Guide; National Youth Policy Institute: Sejong, Korea, 2017.

In response of your comment, we separated the measures to dependent variables, independent variables, and socio-demographic ones as below:

p.3)

Depressive symptoms and aggression were dependent variables. Depressive symptoms were measured using ten items from the Depression Scales of the Korean Mental Diagnosis Test [36]. Aggression was measured using six items from the Aggressive Behavior Rating Scale [37]. Responses to each question were rated on a 4-point Likert scale, ranging from 1 (strongly disagree) to 4 (strongly agree). A higher score indicated a higher level of depressive symptoms or aggression. The Cronbach alpha coefficients of the items for depressive symptoms were 0.902 (wave 5), 0.892 (wave 6), 0.893 (wave 7), and for aggression: 0.816 (wave 5), 0.811 (wave 6), 0.822 (wave 7). Independent variables were abuse and neglect, which were measured with four items (α = 0.857 in wave 5) from the Abusive Parenting Scale and four items (α = 0.73 in wave 5) from the Neglectful Parenting Scale, respectively, rated on a 4-point Likert scale [38]. The higher the value, the higher the level of abuse or neglect.

We included household income, health status, satisfaction with academic performance, and peer relationships as confounders since prior literature showed a relationship between socio-demographic characteristics and psychological and behavioral problems in adolescents [39,40]. Annual household income was assessed by ranking from “10 million won or less = 1,” “between 10 million and 20 million won = 2,” to “up to over 100 million won = 11,” with 11 classifications in total. Health status was measured with the response to the question, “How do you feel about your health compared to your peers?” on a 4-point Likert scale (“very unhealthy” = 1 to “very healthy” = 4). Satisfaction with academic performance was measured with the response to the question, “How satisfied are you with your academic performance?” on a 4-point Likert scale (“very unsatisfied” = 1 to “very satisfied” = 4). Peer relationships were assessed with five items from the School Life Adaptation Scale – Peer Relationships responding on a 4-point Likert scale [41]. The higher the value, the better the peer relationships. Gender was coded as “male = 0,” and “female = 1.”

  1. The data analyses section might report all analyses conducted and explain / relate them to the hypotheses / aims.

Response

In response of your comment, we revised the data analysis description as below:

p.4)

We calculated the mean values ​​of depressive symptoms, aggression, abuse, neglect, and socio-demographic status (household income, health status, satisfaction with academic performance, peer relationships, and age) of the study subjects through descriptive statistics. In addition, t-test was conducted to determine if there is a significant difference between the means of male and female group. We used the latent growth model (LGM) to examine the trajectories of depressive symptoms and aggression. The LGM estimates the magnitude of change at the group and individual levels using longitudinal data. In the first step, we conducted a conditional LGM to examine the effects of abuse and neglect on the trajectories of depressive symptoms and aggression. In the second step, we included socio-demographic variables in the analysis to control their effects on depressive symptoms and aggression. We used data from the fifth wave as explanatory variables, assuming that they were time-invariant covariates [42]. Next, we conducted a multi-group comparison analysis to investigate gender differences in the association between abuse, neglect, depressive symptoms, and aggression. We used the statistical software packages AMOS 23 and SPSS 22 for the application of the LGM.

  1. Results

6.1. First, authors should study the differences in socio-demographic data. Now, there are combined with the outcomes. I think it would add clarity for the reader if the analyses are conducted step by step.

Response

In response of your comment, we conducted the analysis step by step. In the fist step, we analyzed the association between abuse, neglect, depressive symptoms, and aggression. In the second step, we included socio-demographic variables in the analysis. Table 2 and 3 represent the analysis results of step 1 and 2.

6.2. Regarding the amount of comparison conducted, I think the authors should conduct a MANCOVA, and/or correct this analysis for multiple testing.

Response

We thank for your suggestion. However, we think MANCOVA is a bit beyond the scope of research purposes.

6.3. The Figures should be corrected I guess. The outcomes should be depression W5, but then W6 and W7 (not W5 in each outcomes, not?)

Response

In response of your comment, we corrected W5, W6, and W7 of depressive symptoms and aggression in figure 1 and 2. 

  1. Discussion

In this section, authors should avoid technical terms. Slope should be “translated” in usual words, to help the reader understand more the impact of the results in the understanding of the development or occurrence of psychopathologies during adolescence.

Response

In response of your comment, we translated the meaning of slope as below:

p.9)

-Abuse increased the initial value of depressive symptoms in adolescence, while lowering the increase rate (slope) of depressive symptoms over time.

-Abuse increased the initial value of aggression and influenced girls more, but decreased the increase rate (slope) of aggression over time in girls.

In response of your comment, we stated the results of the socio-demographic variable related results in the result section. Also, we added interpretation of the socio-demographic variable related results which have significant gender differences in the discussion section as below:

p.5) Results

The better the health status, the lower the initial value of depressive symptoms, and the effect was greater in girls (b = -0.245, p < 0.001) than in boys (b = -0.139, p < 0.001). The higher the satisfaction of academic performance, the lower the initial value of depressive symptoms in both boys (b = -0.092, p < 0.001) and girls (b = -0.107, p < 0.001). The better the peer relationships, the lower the initial value of depressive symptoms for both boys (b = -0.301, p < 0.001) and girls (b = -0.421, p < 0.001). Peer relationships increased the depressive symptoms slope for girls (b = 0.1, p < 0.001), but it did not affect the depressive symptoms slope for boys.

p.9-10) Discussion

Health status reduced depressive symptoms more in girls than in boys. The health-related differences between the genders are remarkable in adolescence [52]. For example, with the onset of puberty, girls are more dissatisfied with their health and receive medical care more often; Girls experience more psychosomatic disorders than boys. Therefore, physical health can be more closely related to the depressive symptoms of girls. Peer relationships decreased the initial values ​​of depressive symptoms in both boys and girls. It increased the increase rate of depressive symptoms over time (slope) in girls, but did not affect the boys' slope. The girls tend to be more interpersonal-oriented than boys in adolescence [45]. Good friendships increase a girl's depressive symptoms in early adolescence. However, over time they can compare their appearance, abilities, and home environments to their peers. This social comparison may occur more frequently among close friends, and it may lead to the increase of depressive symptoms over time [53].

  1. Minor point

Some sections (e.g., 2.1) are in italic which does not have to be the case.

Response

In response of your comment, we revised the incorrect italic format.